# TeamCraft: A Benchmark for Embodied Multi-Agent Systems in Minecraft

## Abstract

Complex 3D environments replete with dynamic interactions among multiple agents and objects are essential for the development of embodied intelligent agents. To facilitate research on Multi-Agent (MA) systems, we introduce *TeamCraft*, a challenging MA benchmark based on the Minecraft game. Instead of the abstract vector inputs commonly provided to agents in MA systems research, *TeamCraft* provides agents with multi-modal task specifications and observations. Given the three-orthographic-view graph of the environment along with language instructions, the agents must efficiently collaborate to complete assigned tasks. Such multi-modal inputs pose a higher level of difficulty, since agents must generalize across diverse object and background imagery, different numbers of agents, a wide range of tasks, etc. Our planner-generated dataset includes various tasks, such as building construction, smelting, and farming, with a total of 70,000 procedurally-generated demonstrations that feature over 50 objects across a wide variety of scenes. We test the generalization abilities of several baseline Vision-Language Model (VLM) multi-agent control strategies in centralized and decentralized settings. The *TeamCraft* platform and dataset are made publicly available at: https://github.com/teamcraft-bench/teamcraft.

## 1 Introduction

In an open-ended world, multiple autonomous agents with diverse skill sets should collaborate to efficiently perform a broad spectrum of tasks. However, research aimed at developing agents capable of efficiently performing intricate tasks within complex, embodied Multi-Agent (MA) environments remains relatively limited. In particular, the autonomous agent research community has primarily focused on navigation and object-based interactions in vision-language task-planning.

One research avenue endeavors to establish MA methodologies within 2D environments employing solely vector inputs (Leibo et al., 2021; Suarez et al., 2021). However, such inputs suffer from limited realism and are characterized by an inherent scarcity of comprehensive information. Concurrently, another research avenue focuses on the creation of singular multi-task agents with the ability to proficiently undertake a diverse range of tasks within domains encompassing both gaming and robotics (Wang et al., 2023b;a; Ahn et al., 2022; Huang et al., 2022b;a). However, when considering the elaborate interactions and uncertainties that arise in MA systems, the endeavor of formulating multi-task agents within MA settings is decidedly more formidable and challenging.

To foster advancements in this domain, we have developed a comprehensive benchmark tailored to MA embodied systems, dubbed *TeamCraft*. It utilizes the acclaimed Minecraft game as an experimental platform and is targeted at confronting the elaborate dynamics of MA interactions. The benchmark encompasses the design of four multi-modal task categories: building construction, ground clearing, farming, and object acquisition. Within the cooperative tasks, each assignment necessitates consideration of fellow agents, spanning factors such as spatial positioning, inventory holdings, skill differentials, and initial vitality. Such nuanced assessments require divergent role allocation and task strategies during the planning phase. The collaborative actions unfolding during the execution phase encompass resource sharing and joint pursuit.

The *TeamCraft* dataset encompasses fundamental skills and tasks, meticulously orchestrated by hand-designed planners. We introduce two alternative baseline models, both trained on the *TeamCraft*

dataset, that validate the efficacy of the generated data. The first model, MA-GPT-4o, employs a Multi-modal Large Language Model (MLLM) as the planner to generate subgoals that guide individual agents. The second model, MA-LLAVA, comprehensively encodes input facets and subsequently fuses embeddings through an attention mechanism, culminating in the prediction of ultimate actions. Our experimental findings demonstrate that both baseline models achieve competence across a subset of tasks.

In a nutshell, the primary contributions of this paper to the MA research community are as follows:

1. *TeamCraft*, a new embodied multi-modal multi-agent benchmark encompassing complex tasks challenging multi-agent systems in a wide variety of generalization scenarios.

2. Novel applications of the GPT-4o and LLAVA models tailored to multi-agent scenarios.

## 2 RELATED WORK

**Environments for Multi-Agent Reinforcement Learning (MARL)**: The recent success of MARL methods (Lowe et al., 2020; Yu et al., 2021; Long et al., 2020; 2024) has garnered attention, as these methods explore cooperation and competence behaviors among agents. These methodologies have been developed and tested on prominent platforms. However, many of these platforms involve 2D environments (Leibo et al., 2021; Suarez et al., 2021; Mordatch & Abbeel, 2017; Vinyals et al., 2019) and rely solely on vector observations. This limited scope poses challenges in terms of extending applicability to real-world scenarios.

**Environments based on Minecraft**: Minecraft games have fostered the development of embodied AI methods. Initially, Malmo (Johnson et al., 2016) marked the advent of a Gym-style API tailored to Minecraft. This endeavor paved the way for subsequent developments, such as MineRL (Guss et al., 2019) and MineDojo (Fan et al., 2022), which augmented the dataset and introduced a suite of benchmarking tasks. However, the focus of these benchmarks predominantly centers around single-agent tasks, with limited exploration of multi-agent scenarios in Minecraft. Despite their contributions, they remain devoid of multi-agent tasks. By contrast, *TeamCraft* concentrates exclusively on the multi-agent setting. This distinctively sets it apart from all preceding Minecraft benchmarks.

**Embodied agents in MA systems**: Within the embodied multi-agent setting, several researchers have employed the AI2-THOR environment (Kolve et al., 2022). Jain et al. (2019) delved into the communication dynamics that enhance collaboration between two agents. Tan et al. (2020b) and Liu et al. (2022a) propounded the efficient exploration of environments as a central task for agents. Meanwhile, Liu et al. (2022b) introduced a model that dynamically decomposes tasks among different agents, enabling dynamic task allocation. It is noteworthy, however, that the task propositions thus far have primarily revolved around navigation subject to environmental constraints. However, Minecraft is a multidimensional, visually immersive realm characterized by procedurally generated landscapes and extraordinarily versatile game mechanics supporting an extensive spectrum of activities. This provides rich environments ripe for intricate collaborations and the emergence of competence.

**Comparison:** Table 1 compares *TeamCraft* with prior benchmarks.

## 3 TEAMCRAFT BENCHMARK

Existing benchmarks in MA systems research are founded on state or voxel-based observation in a controlled, closed environment. Actions and task types are also limited by the environments. *TeamCraft* advances the state-of-the-art in multi-agent benchmarks by exploiting the dynamic and open-ended Minecraft environment, offering 1) high-quality RGB first-person perspective observations on top of the traditional voxel-based and state-based observations, 2) the ability to benchmark on existing MA cooperation tasks and define custom tasks with a variety of interactions, 3) the ability to control multiple agents in an open world to perform open-ended 3D tasks, in both centralized or decentralized settings, 4) the capacity to execute hundreds of actions individually for multiple agents that expand all possible task spaces with high-level, abstract language input, and 5) the ability to provide expansive visual diversity in tools, blocks, entities, and richly-detailed backgrounds.

Table 1: Comparison of *TeamCraft* and other benchmarks. *TeamCraft* features RGB image and language inputs for multi-agents control with a large number of widely-varied demonstrations in Minecraft. The columns refer to the following features: **RGB:** Real-time first-person perspective RGB images are provided to agents and serve as observations. **Language:** Task goals are specified by human language instruction. **3D:** Task requires agents to have perception and be able to interact with the 3D world (i.e., movement in 3D, objects interacted with have 3D relations). Note: "Obs" denotes only support of 3D observation, no movement or action in 3D. **Allocation:** Multiple tasks must be dynamically allocated to multiple agents to obtain maximum benefit. Agents must use visual perception to understand other agents' states and make decisions to increase efficiency. **Multi-Agents:** Multiple agents can be present in a single experiment. **(De)centralized:** Agents can be operated separately in both centralized and decentralized settings. **Tool Use:** Completing tasks necessitates the use of specific tools by the agents, or using various tools results in different task efficiencies. **Interaction:** Agents must manipulate or engage with different items or environmental elements or objects to achieve certain goals with irreversible actions. **Generalization**: Standardized generalization across a diversity of goals, objects, backgrounds, and inventories.

| Benchmark | RGB | Language | 3D | Allocation | Multi-Agents | (De)centralized | Tool Use | Interaction | Generalization |
|---|---|---|---|---|---|---|---|---|---|
| Alfred (Shridhar et al., 2020) | ✓ | ✓ | Obs | ✗ | ✗ | ✗ | ✓ | ✓ | 100,000+ |
| DialFRED (Gao et al., 2022) | ✓ | ✓ | Obs | ✗ | ✗ | ✗ | ✓ | ✓ | 53,000+ |
| MultiagentEQ (Tan et al., 2020a) | ✓ | ✓ | Obs | ✗ | ✓ | ✗ | ✓ | ✓ | ✗ |
| EmbodiedMA (Liu et al., 2022b) | ✓ | ✓ | Obs | ✓ | ✓ | ✓ | ✗ | ✗ | ✗ |
| Cordial Sync (Jain et al., 2020) | ✓ | ✓ | Obs w/ Action | ✓ | ✓ | ✓ | ✗ | ✓ | ✗ |
| MineLand (Yu et al., 2024) | ✗ | ✓ | ✓ | ✓ | ✓ | ✗ | ✓ | ✓ | 6,000+ |
| MindAgent (Gong et al., 2023) | ✗ | ✓ | Obs | ✓ | ✓ | ✗ | ✓ | ✓ | 100,000+ |
| Creative Agents (Zhang et al., 2023) | ✓ | ✓ | ✓ | N/A | ✗ | N/A | ✓ | ✓ | ✗ |
| MineDojo (Fan et al., 2022) | ✓ | ✓ | ✓ | N/A | ✗ | N/A | ✓ | ✓ | 1,000+ |
| Overcooked-AI (Carroll et al., 2020) | ✗ | ✗ | Obs | ✓ | ✓ | N/A | ✗ | ✓ | ✗ |
| Watch&Help (Puig et al., 2021) | ✗ | ✗ | Obs | ✓ | ✓ | ✗ | ✗ | ✓ | ✗ |
| Too many cooks (Wang et al., 2020) | ✗ | ✗ | ✗ | ✓ | ✓ | ✓ | ✗ | ✓ | ✗ |
| SQA3D (Ma et al., 2023) | ✓ | ✓ | ✓ | ✗ | ✗ | N/A | ✗ | ✗ | 40,000+ |
| OpenEQA (Majumdar et al., 2024) | ✓ | ✓ | Obs | ✗ | ✗ | N/A | ✗ | ✗ | 2,000+ |
| AlexaArena (Gao et al., 2023) | ✓ | ✓ | Obs | ✗ | ✗ | N/A | ✓ | ✓ | ✗ |
| ***TeamCraft*** | ✓ | ✓ | ✓ | ✓ | ✓ | ✓ | ✓ | ✓ | 70,000+ |

## 3.1 SIMULATION ENVIRONMENT

*TeamCraft* utilizes Minecraft as its foundational environment, offering a complex, open-world setting for multi-agent interactions. The environment features procedurally generated tasks, visually rich changes, web-scale knowledge, and diverse cooperation strategies among agents. Each agent is individually controlled via the Mineflayer[1] interface, which provides low-level API functionalities for bots to interact with the environment. *TeamCraft* utilizes Mineflayer's APIs to 1) translate high-level actions into low-level commands through nested API calls, 2) generate both first-person and third-person RGB image perspectives, 3) enable Gym-like interactions across four tasks that challenge visual perception, spatial reasoning, and multi-agent task planning, and 4) support multiple agents. This framework facilitates the execution of intricate commands via self-explanatory high-level actions, allowing agents to collaboratively complete sophisticated tasks. Figure 1 illustrates the platform architecture.

## 3.2 OBSERVATION AND ACTIONS

*TeamCraft* captures a wide array of observational data to ensure that agents have a comprehensive understanding of their environment.

**Visual:** It provides $640 \times 480$ resolution images from a first-person perspective for agents before each time step. It also provides orthographic projections images for task specifications. Images are also illustrated in each task description.

**Agent inventory:** It provides detailed reporting about each agent's inventory.

The action space mainly involves high-level self-explanatory skills such as *obtainBlock* and *farmWork*. We provide 8 such atomic actions. Most actions take three input parameters, including 1) agent name such as *bot1*, as the action-executing entity, 2) item name such as *dirt*, which is strongly associated

---

[1] https://github.com/PrismarineJS/mineflayer

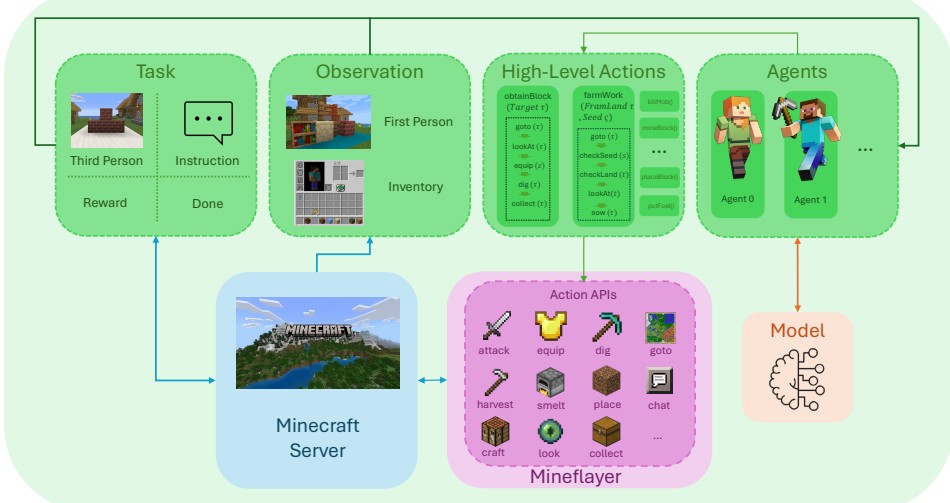

Figure 1: *TeamCraft* platform architecture consists of three main components: 1) a Minecraft server that hosts the game as an online platform, 2) Mineflayer, which serves as the interface for creating and controlling bots in the Minecraft server, and 3) a Gym-like environment that defines tasks, provides RGB and inventory observations, and allows models to control multiple agents through high-level actions.

with the task goal or the agent's inventory, 3) a vector indicating the position of the target on the test field. A complete list of the atomic actions are described in Appendix C.

### 3.3 CENTRALIZED AND DECENTRALIZED AGENTS

We have implemented two different categories of agents: centralized agents and decentralized agents.

**Centralized agents:** These agents are given complete observational access to the environment, including the first person view, action history, and inventory information of all the agents. Based on these comprehensive data, the model generates the actions for all agents simultaneously. This approach leverages the full scope of information available in the environment to coordinate and optimize the actions of all the agents collectively.

**Decentralized agents:** These agents do not receive information about other agents except for the initial environment settings, which may include some inventory details of other agents, and the task description. The model generates actions solely for each individual agent based on the agent's own limited view. This setting simulates a more realistic scenario where agents operate independently with restricted information, focusing on their own observations and actions absent of any centralized coordination.

### 3.4 TASK DESIGN

*TeamCraft* introduces a variety of complex and interactive tasks that challenge the agents' capabilities in planning, coordination, and execution within a collaborative and dynamic environment. Each task is designed to test different facets of MA interaction, including communication strategies, role distribution, real-time decision-making, and adaptability to changing environments. Tasks require capabilities in visual observation understanding, agent status intercepting, action capability understanding, language prompt understanding, continuous state understanding, and task action sequence planning. Here, we detail the specific tasks included in the *TeamCraft* benchmark. Task examples are shown in Figure 2.

**Building:** This task requires agents to collaboratively erect a structure based on a provided three orthographic views blueprint (front, side, and top). Each agent possesses a unique inventory of building blocks necessary for the construction. The task requires agents not only to understand their

| | Building | Clearing | Farming | Farming | Smelting | Smelting |
|---|---|---|---|---|---|---|
| Scenes | village | snow_mountain | village | swamp | ice_on_water | desert_villege |
| Base | cyan_concrete | gold_block | hay_block | obsidian | oak_wood | glass |
| Goal | Build 1x2x4 building | Clean 3D building | Potato *3 | wheat *4 | cooked_mutton *1 | smooth_quartz *2 |
| Object | [dirt, wool, fence sandstone, sponge] | [grass_block, dirt birch_log, bookshelf,] | - | - | [birch_planks, sheep] | [oak_planks, quartz_block] |
| Agent | 3 | 3 | 2 | 2 | 3 | 2 |
| Inventory | [dirt, wool, fence sandstone, sponge, log, stone, sand] | [stone_axe, stone_sword, stone_sword] | [carrot, beetroot] | [wheat_seeds, carrot, potato] | [iron_pickaxe, iron_axe, iron_sword] | [iron_pickaxe, iron_axe] |
| Demonstration | | | | | | |

Figure 2: Examples of the four tasks. We introduce 7 scenes featuring over 40 blocks and objects, which are arranged into more than 40,000 unique placement configurations. A detailed distribution is provided in Appendix H.

individual capabilities and inventories, but also to plan their movements and actions in coordination with other agents so as to efficiently construct the building on a designated $5 \times 5$ foundation.

**Clearing:** This task challenges agents to remove all blocks from a specified $6 \times 6$ area. Agents must employ appropriate tools to break the blocks, which vary in durability, thereby requiring multiple interactions for complete removal. The use of correct tools can dramatically reduce the time required to remove blocks (up to $3\times$ speedup). The agents must manage their tool assignments to optimize block-breaking efficiency such that the time steps needed for one task can be minimized. Strategic coordination is essential in this task as agents need to dynamically decide which blocks to target based on their current tools and help each other minimize the overall time taken to clear the area.

**Farming:** This task is designed to simulate agricultural activities, where agents must sow and harvest crops. Agents are required to plant seeds on designated farmland plots and observe plantings until the crops reach maturity. Each crop has several growth stages from Level 0 (newly planted) to Level 7 (fully grown), and agents must identify when crops are ready to be harvested. The challenge lies in dynamically allocating tasks among agents based on their positions, available seeds, and the maturity of different crops. Effective task distribution and coordinated actions ensure maximum yield and efficiency. For example, some agents can sow while others are planting, and they should stop when their total crop yield is satisfactory.

**Smelting:** This task requires agents to obtain items processed using furnaces by gathering materials and coordinating actions. Agents collect resources from the environment—by harvesting blocks or killing mobs—and place them, or existing inventory items, into furnaces as smelting inputs. The output will be the final goal item that can be categorized as food or item, where food can be "cooked beef", "cooked porkchop", or "baked potato", and item can be "glass" or "gold ingot" by smelting sand or gold ore, respectively. Agents must also gather fuel (e.g., coal or lava buckets), with each furnace accepting only one type of fuel. Furnaces are placed near the playground center (one or two per task) and automatically smelt when supplied with fuel and items. Agents must use the provided tools, communicate effectively, and assign tasks efficiently due to dependencies in the smelting process.

## 3.5 MULTI-MODAL PROMPT

For each task, the benchmark provides a multi-modal prompt consisting of both a set of orthographic projections (i.e. top, left, front views) and a language instruction for task specification. For the building task, the images depict the target structure. For tasks such as clearing, farming, and smelting, the images will show the initial state of the environment. The language instruction will specify the goal: for building, it will be "build a structure"; for clearing, "break the blocks on the platform"; for farming, "harvest a specific number of crops"; and for smelting, "smelt a specific number of items". The detailed prompt examples are shown in Figure 3

## 3.6 DIVERSITY

The design of these tasks incorporates several layers of complexity to test and develop robust multi-agent systems capable of operating in diverse and unpredictable environments. Table 2 shows the

|  |  | Building | Clearing | Farming | Smelting |
|---|---|---|---|---|---|
| System Prompt | Three orthographic views | Top  Front  Side | Top  Front  Side | Top  Front  Side | Top  Front  Side |
|  | Language Instruction | Three bots need to build a building on the platform. bot1 has 5 bricks. bot1 has 2 sea_lantern. bot1 has 3 iron_ore ... bot3 has 1 brick... Write the actions for bot1, bot2 and bot3 based on this given observation. | Three bots need to break everything on the platform. bot1 has a stone_axe..., bot3 has a stone_axe. Write the actions for bot1, bot2, bot3 based on this given observation. | Two bots need to grow on the platform. The goal is to get 4 carrot. bot1 has 3 carrot. bot1 has 1 potato. bot2 has 3 carrot. bot2 has 2 beetroot. Write the actions for bot1, bot2 based on this given observation. | Three bots need to craft 3 smooth_sandstone. here are the introductions: Cooking Food: 1. To cook a 'cooked_beef', I need 'beef'. To get 'beef', I need to kill a 'cow' or a 'mushroom'. 2. To cook a 'cooked_porkchop' ... bot1 has 1 beef ... bot3 has 1 iron_shovel. Write the actions for bot1, bot2 and bot3 based on this given observation. |
| Observation | First Person View | bot1  bot2  bot3 | bot1  bot2  bot3 | bot1  bot2 | bot1  bot2  bot3 |
|  | Inventory Information | bot1 has 5 bricks. bot1 has 2 sea_lantern... | bot1 has a stone_axe...bot3 has a stone_axe... | bot1 has 3 carrot. bot1 has 1 potato. bot2 has 3 carrot. bot2 has 2 beetroot... | bot1 has 1 beef ... bot3 has 1 iron_shovel |
|  | Action | placeItem(bot1, 'bricks', new Vec3(-1,0,-1))  placeItem(bot2, 'oak_planks', new Vec3(0,0,0))  placeItem(bot3, 'iron_ore', new Vec3(0,0,-1)) | mineBlock(bot1, new Vec3(-1,0,1))  mineBlock(bot2, new Vec3(-2,0,0))  mineBlock(bot3, new Vec3(-1,1,1)) | farm_work(bot1, new Vec3(-1,-1,1), 'sow', 'carrot')  farm_work(bot2, new Vec3(-1,-1,-2), 'sow', 'carrot') | putItemFurnace(bot1, 'sandstone', new Vec3(0,0,-1))",  "obtainBlock(bot2, new Vec3(2,0,0))",  "obtainBlock(bot3, new Vec3(1,0,-3)) |

Figure 3: Multi-modal prompts are provided for all tasks. The system prompt includes both the three orthographic views and specific language instructions. Observations consist of first-person views from different agents, along with agent-specific information.

statistics and variants for each task. Appendix E demonstrates a sample of the visual diversity included.

**Object diversity:** More than 30 3D objects are used as the target item or resource in tasks. Objects, such as a fence, an anvil, or a stone block, have different shapes and different textures, such as pink wool and dirty blocks. Farm crops will have different visual appearances during growth so that the agent can determine their growth stages from observations. The smelting task requires agents to obtain different resources, such as killing different mods that have different shape, size, and orientation, such as a chicken, rabbit, or pig.

**Inventory diversity:** Each agent's inventory might include essential items mixed with non-essential ones (i.e., distractors), realistically simulating scenarios where agents must choose the right materials for specific tasks while managing inventory constraints. Agents are also provided with random tools at the beginning of each task, which are critical for efficient action execution. Possessing the proper tools impacts task efficiency in the clearing task and can lead to action failure in smelting when collecting blocks.

**Scene diversity:** More than 10 scenes are included in the tasks, covering biomes such as village, mountain, forest, swamp, desert, etc. The task interaction area (e.g., the $5 \times 5$ area for building construction) are spawned in a random position of the scene to ensure visual diversity. Tasks take

Table 2: Task variants and dataset statistics

|  | Building | Clearing | Farming | Smelting |
|---|---|---|---|---|
| # Action Sequences | $2 - 6$ | $2 - 9$ | $2 - 7$ | $2 - 8$ |
| # Agents | $2 - 3$ | $2 - 3$ | $2 - 3$ | $2 - 3$ |
| # Tools | – | $1 - 4$ | – | $1 - 4$ |
| # Scenes | 6 | 5 | 4 | 5 |
| # Base Types | 10 | 11 | 9 | 11 |
| # Furnaces | – | – | – | $1 - 2$ |
| # Target Types | 19 | 16 | 3 | 13 |
| # Target Counts | $5 - 12$ | $4 - 9$ | $2 - 14$ | $1 - 4$ |
| # Fuel Types | – | – | – | 12 |
| # Resource Types | – | – | – | 20 |
| # Dimensional Shapes | 2 | 2 | 2 | 1 |
| # Placement Shapes | 7715 | 12724 | 13188 | 8885 |
| # Total Demonstrations | 14998 | 14641 | 14815 | 10803 |
| # Test Set | 50 | 50 | 50 | 50 |
| # Generalization Set | 200 | 200 | 150 | 200 |

place on grounds with diverse textured bases such as glass, concrete, and quartz. Certain tasks may involve additional complexity, such as farmland intermixed with non-plantable blocks.

**Goal diversity:** Goals vary between tasks. For the place and construction task, we introduce different block placement shapes; e.g., a $2 \times 4 \times 2$ tower with top right intentionally not occupied. We categorized those shapes into different dimensionalities; e.g., 2D (all blocks are at the same level) or 3D (some blocks are on the top of others). For the farming task, the total target corp type and counts are randomized. For the smelting task, the target object is randomized from various food or processed items, and the fuel for smelting is also randomized.

**Task diversity:** Each task requires achieving a varying number of goal targets, determined by the randomly assigned number of agents per task, which range from two to four. This variability challenges the agents' flexibility and adaptability in coordination and task execution. Additionally, differing task requirements lead to varying numbers of actions necessary for optimal task completion.

### 3.7 EXPERT DEMONSTRATION GENERATION PIPELINE

To create a rich learning environment and effective training dataset for the *TeamCraft* tasks, systematic scenario design and data collection methods are employed, as follows:

**Planner-based scenario design:** Each task scenario is carefully crafted using classical planning algorithms, such as BFS, greedy search, and DFS, that consider all possible interactions within the environment. This includes optimal paths, resource distribution, and agent role assignments based on capabilities and task requirements.

**Trajectory generation:** Using Mineflayer interfaces controlled by heuristic methods such as the Hungarian Algorithm and dynamic programming, the planner orchestrates the agents to execute the task, ensuring that actions are taken optimally. Each step's effectiveness is assessed to guarantee efficient task completion.

**Real-time interaction and feedback:** Agents receive immediate feedback on their actions, which includes success, failure, and updates on environmental states. This real-time data is crucial for adjusting strategies and learning from interactions.

### 3.8 TEST SET AND GENERALIZATION SET

Each task features a test set, where agents are initialized with random position, orientation, and inventory. The rest variables follow the same distribution as the training data. To evaluate specific generalization capabilities of the model, we designed a generalization set for each task with hold-out elements excluded from the training data. We withheld test cases involving four agents, whereas the training demonstrations include only two or three agents. We also introduced one unseen scene and an associated base block type not present during training. In addition to these general hold-outs, we implemented the following task-specific exclusions:

**Building task:** We randomly excluded 8 block placement shapes, defining how target blocks are arranged on the ground. These shapes varied in complexity, containing 5 to 12 blocks in both 2D and 3D configurations. Additionally, we omitted 3 block materials that appeared in the clearing task but not in the building task.

**Clearing task:** We randomly held out 6 block placement shapes with block counts ranging from 4 to 9. We also excluded 3 block materials present in the building task but absent in the clearing task.

**Farming task:** We withheld one crop type, beetroot, that was unseen during training.

**Smelting task:** We excluded four unseen objects from both food and item categories and introduced scenarios with 3 furnaces, as opposed to 1 to 2 furnaces in the training data.

As shown in Table 2, with 50 samples per task for the test set and each generalization set, our benchmark contains a total of 950 test cases.

# 4 EXPERIMENTS

## 4.1 BASELINES

In our experiments, we utilized the pretrained LLaVA-v1.6-Vicuna-7B and LLaVA-v1.6-Vicuna-13B models. We modified the LLaVA architecture by concatenating image embeddings with language embeddings to handle multiple images. All models were pretrained for 3 epochs. The model's input includes both the system prompt and the agent's observation. We trained a unified model for all tasks in both the centralized and decentralized settings.

In the centralized setting, the observation consists of first-person views, previous actions, and the information of all agents.

In the decentralized setting, the observation includes the first-person view, previous actions, and information of only the specific agent.

**GPT-4o:** For the GPT-4o method, we employed a one-shot learning approach. The prompt provided to the model includes a single successful demonstration of the task from the training set. Based on this example, we then asked the GPT-4o model to generate the actions for agents in response to new observations. This approach leverages the model's ability to generalize from a minimal amount of information.

## 4.2 EVALUATION METRICS

We evaluated the performance of the methods based on two key metrics: task success rate and competence percentage.

**Task success rate:** The task success rate is determined by the ratio of the number of completed tasks to the total number of tested tasks. This metric indicates the proportion of test cases that the model can successfully complete from start to finish. A higher success rate reflects the model's ability to consistently achieve the desired outcomes in various scenarios.

**Subgoal success rate:** This metric measures the overall effectiveness of the agents in performing the tasks, considering partial successes and the extent to which the tasks are completed. It is calculated by dividing the number of subgoals accomplished by the total number of subgoals. For the building tasks, subgoals are defined by the number of blocks to be built. For the clearing task, subgoals are defined by number of blocks to be cleared. For the farming task, subgoals are defined as the number of farms to be farmed. For the smelting task, subgoals are defined as the number of target objects to be smelt. The subgoal success rate provides a more granular view of the model's performance, highlighting how well the agents can handle different aspects of the tasks even if they do not fully complete them.

## 4.3 EVALUATION RESULTS

We fine-tuned the LLaVA-Vicuna-7B model on three data scales: one-tenth, one-half, and the full training split, in both the centralized and decentralized settings. The task success rates and subgoal success rate are shown in Table 3 with the task success rate on the left and the subgoal success rate on the right.

Comparing horizontally, the centralized settings generally yielded higher task success rates and subgoal success rate, underscoring the advantage of having comprehensive environmental data available to the decision-making processes. By contrast, the decentralized settings showed a noticeable decline in the performance metrics. Even when trained on the full dataset, the model struggled with complex tasks such as building, which requires intricate coordination among agents and detailed interactions with the environment, including correct material selection and coordination. The limited information flow inherent to decentralized settings clearly hindered the models' ability to develop and execute cohesive strategies effectively.

Another observation was the model's adaptability to out-of-distribution parameters. For instance, tasks under the "Test" category generally had higher subgoal success rate, suggesting the models were more proficient at handling familiar scenarios where environmental variables aligned with expected parameters. However, performance declined in tasks involving "Agents", "Scene", "Material", or

Table 3: Experimental results with the 7B MA-LLaVA model. Test refers to the test set with the same distribution as the training data with randomly initialized position, orientation, and inventory of agents. Shape, material, scene, crop, furnace, and agents refer to the generalization set with the corresponding holdout element.

| Tasks | Condition | Centralized | | | Decentralized | | |
|---|---|---|---|---|---|---|---|
| | | 10% | 50% | 100% | 10% | 50% | 100% |
| Building | Test | 0.00 (12.4) | 0.38 (76.7) | 0.42 (81.5) | 0.00 (18.1) | 0.00 (28.7) | 0.00 (38.0) |
| | Shape | 0.00 (12.1) | 0.20 (67.5) | 0.30 (75.5) | 0.00 (15.7) | 0.00 (25.6) | 0.00 (40.1) |
| | Material | 0.00 (13.4) | 0.18 (64.0) | 0.30 (74.2) | 0.00 (13.6) | 0.00 (20.4) | 0.00 (34.0) |
| | Scene | 0.00 (14.7) | 0.36 (72.8) | 0.40 (82.6) | 0.00 (15.6) | 0.00 (20.6) | 0.00 (36.0) |
| | Agents | 0.00 (17.6) | 0.02 (50.3) | 0.02 (57.2) | 0.00 (11.5) | 0.00 (20.1) | 0.00 (14.0) |
| Clearing | Test | 0.00 (13.0) | 0.08 (43.4) | 0.64 (91.2) | 0.00 (45.4) | 0.02 (34.9) | 0.20 (68.0) |
| | Shape | 0.00 (09.0) | 0.08 (34.4) | 0.56 (90.9) | 0.00 (46.6) | 0.02 (27.1) | 0.16 (74.0) |
| | Material | 0.00 (10.0) | 0.12 (45.6) | 0.56 (90.6) | 0.00 (48.9) | 0.00 (22.1) | 0.16 (67.0) |
| | Scene | 0.00 (11.3) | 0.10 (43.8) | 0.58 (92.3) | 0.00 (41.3) | 0.04 (37.4) | 0.10 (64.0) |
| | Agents | 0.00 (15.5) | 0.14 (63.7) | 0.36 (81.3) | 0.02 (50.2) | 0.02 (54.0) | 0.12 (60.0) |
| Farming | Test | 0.14 (43.1) | 0.34 (60.7) | 0.36 (63.8) | 0.02 (07.4) | 0.02 (13.8) | 0.00 (09.0) |
| | Crop | 0.00 (00.0) | 0.00 (00.0) | 0.00 (00.0) | 0.00 (00.0) | 0.00 (00.0) | 0.00 (00.0) |
| | Scene | 0.16 (38.9) | 0.34 (65.1) | 0.38 (66.9) | 0.00 (05.0) | 0.00 (10.5) | 0.02 (07.3) |
| | Agents | 0.02 (17.5) | 0.18 (60.8) | 0.38 (68.4) | 0.00 (07.9) | 0.00 (10.5) | 0.04 (27.0) |
| Smelting | Test | 0.06 (17.4) | 0.20 (36.0) | 0.24 (28.0) | 0.08 (13.3) | 0.08 (09.5) | 0.16 (29.1) |
| | Goal | 0.08 (20.9) | 0.04 (07.5) | 0.00 (00.0) | 0.08 (17.3) | 0.00 (00.0) | 0.00 (00.0) |
| | Furnace | 0.10 (28.3) | 0.10 (20.5) | 0.18 (20.0) | 0.06 (07.0) | 0.06 (06.0) | 0.06 (15.6) |
| | Scene | 0.08 (19.1) | 0.14 (27.8) | 0.18 (23.0) | 0.08 (18.6) | 0.14 (19.8) | 0.12 (27.8) |
| | Agents | 0.00 (15.1) | 0.02 (23.9) | 0.06 (13.1) | 0.04 (04.8) | 0.00 (01.6) | 0.02 (28.0) |

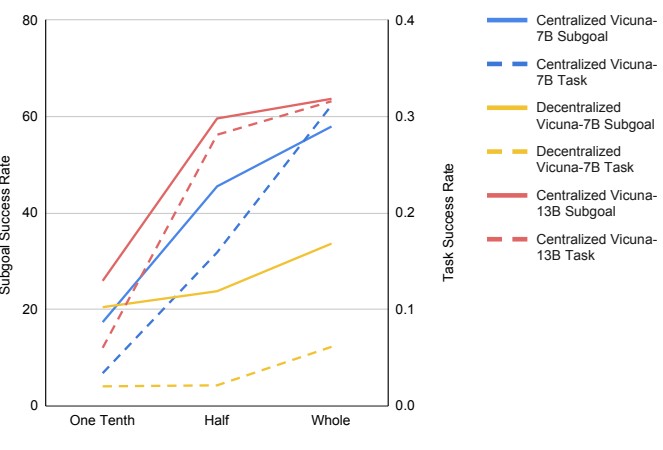

Figure 4: Models performance with different scale of training data.

"Goal" conditions, where unpredictable elements affected task dynamics. Notably, all models failed when dealing with new crops in the farming task, indicating a potential area for improvement in enhancing model robustness and adaptability to unseen scenarios.

We show the scaling law in Figure 4. As the training data increased, we observed significant improvements in both subgoal success rate and task success rates across both settings, highlighting the importance of our dataset in achieving better performance.

We also fine-tuned the LLaVA-Vicuna-13B model under centralized settings and compared it to the fine-tuned LLaVA-Vicuna-7B and GPT-4o models, as shown in Table 4 with the task success rate on the left and subgoal success rate on the right. The results show that the LLaVA-Vicuna-13B model

Table 4: Ablations on the base model under the centralized setting

| Tasks | Condition | Vicuna-7B | Vicuna-13B | GPT-4o |
|---|---|---|---|---|
| Building | Test | 0.42 (81.5) | 0.48 (79.2) | 0.00 (07.5) |
| | Shape | 0.30 (75.5) | 0.26 (68.6) | 0.00 (08.1) |
| | Material | 0.30 (74.2) | 0.08 (63.2) | 0.00 (07.4) |
| | Scene | 0.40 (82.6) | 0.48 (83.3) | 0.00 (07.0) |
| | Agents | 0.02 (57.2) | 0.04 (58.5) | 0.00 (00.0) |
| Clearing | Test | 0.64 (91.2) | 0.64 (93.7) | 0.00 (3.0) |
| | Shape | 0.56 (90.9) | 0.78 (96.4) | 0.00 (3.5) |
| | Material | 0.56 (90.6) | 0.56 (91.7) | 0.00 (1.2) |
| | Scene | 0.58 (92.3) | 0.48 (90.4) | 0.00 (5.7) |
| | Agents | 0.36 (81.3) | 0.16 (76.5) | 0.00 (0.0) |
| Farming | Test | 0.36 (63.8) | 0.46 (72.6) | 0.00 (0.00) |
| | Crop | 0.00 (00.0) | 0.00 (00.0) | 0.00 (0.00) |
| | Scene | 0.38 (66.9) | 0.44 (74.5) | 0.00 (0.00) |
| | Agents | 0.38 (68.4) | 0.36 (71.9) | 0.00 (0.00) |
| Smelting | Test | 0.24 (28.0) | 0.32 (58.5) | 0.02 (2.00) |
| | Goal | 0.00 (00.0) | 0.00 (00.0) | 0.08 (8.00) |
| | Furnace | 0.18 (20.0) | 0.18 (38.3) | 0.00 (0.00) |
| | Scene | 0.18 (23.0) | 0.24 (55.8) | 0.00 (0.00) |
| | Agents | 0.06 (13.1) | 0.04 (36.6) | 0.00 (0.00) |

outperforms both the Vicuna-7B and GPT-4o models. GPT-4o, using a one-shot demonstration, struggled to complete most tasks and achieved a significantly lower subgoal success rate compared to the fine-tuned models, with the exception of a few successes in the smelting task. The smelting task is less reliant on precise coordination since the locations of the stoves are fixed at three positions, and it is possible that agents already have the necessary materials in their bags, eliminating the need to gather resources. This highlights the limitations of Large Language Models (LLMs) in 3D spatial reasoning and emphasizes the difficulty of multi-modal tasks, further underscoring the critical role our dataset can play in advancing performance.

## 5 CONCLUSIONS

The *TeamCraft* benchmark introduced in this paper provides a novel and rich framework for evaluating the capabilities of multi-agent systems situated in complex 3D environments. By incorporating a diverse array of tasks, coupled with dynamic interactions among agents and objects, this benchmark challenges the conventional paradigm of multi-agent research and paves the way for new explorations in embodied intelligence.

The implementation of RGB image and language inputs as opposed to traditional abstract vector inputs has enabled a more realistic simulation of human-like perception and interaction. This setup has effectively demonstrated the necessity and impact of high-level strategic planning and real-time decision-making in a controlled yet challenging environment.

Our experimental results highlight the strengths and limitations of current Vision-Language Models (VLMs) in managing complex, dynamic task environments. While the centralized models exhibited robust performance across most tasks, reflecting their ability to leverage comprehensive environmental data for decision-making, the decentralized models underscored the challenges faced when agents operate with limited information. This dichotomy not only enriches our understanding of agent interaction dynamics but also underscores the critical role of information accessibility in strategic multi-agent environments.

In conclusion, the *TeamCraft* benchmark not only sets a new standard in the study of multi-agent systems but also promises to act as a catalyst for future innovations in this rapidly evolving field.

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

## A  PROMPT EXAMPLES

We include some prompt examples for *TeamCraft*. The information includes task specific requirement and agents' current states.

For the building task, we provide a three orthographic views of the building to accomplish, and we also include the agents inventory information. Here is one example:

*"<image>Two bots need to build a building on the platform. bot1 has 6 coal_ore. bot1 has 3 clay. bot1 has 4 sandstone. bot1 has 3 purple_wool. bot1 has 1 bricks. bot2 has 3 bricks. bot2 has 4 purple_wool. bot2 has 5 coal_ore. bot2 has 2 sandstone. Write the actions for bot1, bot2 based on this given observation."*

For the clearing task, we provide a three orthographic views at initialization that they need to clear out, and we also include the agent's tool information. Here is one example:

*"<image>Two bots need to break everything on the platform. bot1 has a stone_axe. bot2 has a stone_axe. Write the actions for bot1, bot2 based on this given observation."*

For the farming task, we provide a three orthographic views of the farmland and agents inventory information. Here is one example:

*"<image>Three bots need to grow on the platform. The goal is to get 4 carrot. bot1 has 3 carrot. bot1 has 1 potato. bot2 has 3 carrot. bot2 has 2 beetroot. bot3 has 5 carrot. bot3 has 1 wheat_seeds. bot3 has 3 potato. bot3 has 1 beetroot. Write the actions for bot1, bot2 and bot3 based on this given observation."*

For the smelting task, we provide an instruction of how to smelt all objects and agents' inventory information. Here is one example:

*"<image> Three bots need to craft 4 cooked_beef. here are the introductions: Cooking Food: 1. To cook a 'cooked_beef', I need 'beef'. To get 'beef', I need to kill a 'cow' or a 'mushroom'.*

*2. To cook a 'cooked_porkchop', I need 'porkchop'. To get 'porkchop', I need to kill a 'pig'.*

*3. To cook a 'cooked_mutton', I need 'mutton'. To get 'mutton', I need to kill a 'sheep'.*

*4. To cook a 'cooked_chicken', I need 'chicken'. To get 'chicken', I need to kill a 'chicken'.*

*5. To cook a 'cooked_rabbit', I need 'rabbit'. To get 'rabbit', I need to kill a 'rabbit'.*

*6. To cook a 'cooked_cod', I need 'cod'.*

*7. To cook a 'cooked_salmon', I need 'salmon'.*

*8. To cook a 'baked_potato', I need a 'potato'.*

*Crafting Items: 1. To craft a 'gold_ingot', I need 'gold_ore'. To get 'gold_ore', I need to obtain 'gold_ore blocks with a pickaxe.*

*2. To craft an 'iron_ingot', I need 'iron_ore'. To get 'iron_ore', I need to obtain 'iron_ore blocks with a pickaxe.*

*3. To craft 'glass', I need 'red_sand'. To get 'red_sand', I need to obtain 'red_sand'.*

*4. To craft 'smooth_sandstone', I need 'sandstone'. To get 'sandstone', I need to obtain 'sandstone' with a pickaxe.*

*5. To craft 'stone', I need 'cobblestone'. To get 'cobblestone', I need to obtain 'cobblestone' with a pickaxe.*

*Fuel Sources:*

*1. To fuel the furnace, I can use 'coal'. To get 'coal', I need to obtain 'coal_ore'.*

*2. To fuel the furnace, I can use 'lava_bucket', 'coal_block', 'charcoal', .*

*3. To fuel the furnace, I can use 'oak_log', 'birch_log', 'acacia_log', 'spruce_log', 'oak_planks', 'birch_planks', 'acacia_planks', or 'spruce_planks'.*

| Type | Arguments | Description |
|------|-----------|-------------|
| placeItem | BotID, ItemType, Location | BotID places an item of ItemType at the specified 3D Location. |
| mineBlock | BotID, Location | BotID mines a block at the specified 3D Location. |
| farmWork | BotID, Location, Action, ItemType | BotID performs an Action (sow or harvest) on ItemType at the specified 3D Location. |
| obtainBlock | BotID, Location | BotID obtains a block from the specified 3D Location. |
| putFuelFurnace | BotID, ItemType, Location | BotID places an ItemType as fuel into a furnace at the specified 3D Location. |
| putItemFurnace | BotID, ItemType, Location | BotID inserts an ItemType into a furnace at the specified 3D Location. |
| takeOutFurnace | BotID, ItemType, Location | BotID removes an ItemType from a furnace at the specified 3D Location. |
| killMob | BotID, Location | BotID engages and eliminates a mob at the specified 3D Location. |

Table 5: Action space within the *TeamCraft*.

*I can also obtain those blocks. I do not need to get those resource if they already in my inventory.bot1 has 1 beef. bot1 has 1 coal_block. bot1 has 2 iron_axe. bot2 has 3 coal_block. bot2 has 1 iron_pickaxe. bot2 has 1 iron_axe. bot3 has 1 iron_shovel. bot3 has 1 iron_axe. Write the actions for bot1, bot2 and bot3 based on this given observation."*

## B  HIGH LEVEL SKILLS

The action space of agents mainly involves high-level self-explanatory skills such as *obtainBlock* and *farmWork*. We provided 8 such atomic actions. Most actions take three input parameters, including 1) agent name such as *bot1*, as the action executing entity, 2) item name such as *dirt*, which strongly associated with task goal or agent's inventory, 3) a vector indicating the position of the target on the test field.

For example, `obtainBlock(bot1, new Vec3(1, 0, 1))` takes the agent name `bot1` and a 3D vector `(1, 0, 1)` as its arguments. It directs `bot1` to perform multiple actions in Minecraft via APIs provided by Mineflayer. First, it controls `bot1` to `goto` a diggable position for block `(1, 0, 1)`, then has `bot1`'s vision ray cast to the block at `(1, 0, 1)` using the `lookAt` action. Next, it commands `bot1` to `equip` a proper tool that can dig the block at `(1, 0, 1)` most efficiently, and then instructs `bot1` to dig the target block. Once the target block has been mined, `bot1` will `goto` the position where the block item dropped and collect it.

Similarly, `farmWork(bot2, "sow", "potato", new Vec3(2, 0, 4))` takes the agent name `bot2`, action type `"sow"` (as opposed to `"harvest"`), crop seed item `"potato"`, and a 3D vector `(2, 0, 4)` as its arguments. It directs `bot2` to `goto` a placeable position for farmland at `(2, 0, 4)`, then `check` if the seed is a valid item—that is, a crop seed available within `bot2`'s inventory. It then `checks` if the farmland at `(2, 0, 4)` is plantable. Finally, it instructs `bot2` to `lookAt` the farmland and `sow` it with the seed `"potato"`.

## C  ATOMIC ACTIONS

Table 5 documents all the atomic actions in our dataset. Atomic functions are JavaScript code instructing Mineflayer via its APIs to control one agent to perform an action in Minecraft.

## D  DETAILED MULTI-MODAL PROMPT

We show a more detailed multi-modal prompt in Figure 5

## E  VISUAL DIVERSITY

Figure 6 illustrates a sample of the visual diversity present in the environment. Each task is visually rich, constructed from a random combination of scene elements, base block types, shapes, goal placements, and target types.

| | | Building | Clearing | Farming | Smelting |
|---|---|---|---|---|---|
| System Prompt | Three orthographic views | Top / Front / Side | Top / Front / Side | Top / Front / Side | Top / Front / Side |
| | Language Instruction | Three bots need to build a building on the platform. bot1 has 5 bricks. bot1 has 2 sea_lantern. bot1 has 3 iron_ore ... bot3 has 1 brick... Write the actions for bot1, bot2 and bot3 based on this given observation. | Three bots need to break everything on the platform. bot1 has a stone_axe..., bot3 has a stone_axe. Write the actions for bot1, bot2, bot3 based on this given observation. | Two bots need to grow on the platform. The goal is to get 4 carrot. bot1 has 3 carrot. bot1 has 1 potato. bot2 has 3 carrot. bot2 has 2 beetroot. Write the actions for bot1, bot2 based on this given observation. | Three bots need to craft 3 smooth_sandstone. here are the introductions: Cooking Food: 1. To cook a 'cooked_beef', I need 'beef'. To get 'beef', I need to kill a 'cow' or a 'mushroom'. 2. To cook a 'cooked_porkchop' ... bot1 has 1 beef ... bot3 has 1 iron_shovel. Write the actions for bot1, bot2 and bot3 based on this given observation. |
| Observation | First Person View | bot1 / bot2 / bot3 | bot1 / bot2 / bot3 | bot1 / bot2 | bot1 / bot2 / bot3 |
| | Inventory Information | bot1 has 5 bricks. bot1 has 2 sea_lantern... | bot1 has a stone_axe...bot3 has a stone_axe... | bot1 has 3 carrot. bot1 has 1 potato. bot2 has 3 carrot. bot2 has 2 beetroot... | bot1 has 1 beef ... bot3 has 1 iron_shovel |
| | Action | placeItem(bot1, 'bricks', new Vec3(-1,0,-1)) placeItem(bot2, 'oak_planks', new Vec3(0,0,0)) placeItem(bot3, 'iron_ore', new Vec3(0,0,-1)) | mineBlock(bot1, new Vec3(-1,0,1)) mineBlock(bot2, new Vec3(-2,0,0)) mineBlock(bot3, new Vec3(-1,1,1)) | farm_work(bot1, new Vec3(-1,-1,1), 'sow', 'carrot') farm_work(bot2, new Vec3(-1,-1,-2), 'sow', 'carrot') | putItemFurnace(bot1, 'sandstone', new Vec3(0,0,-1))", "obtainBlock(bot2, new Vec3(2,0,0))", "obtainBlock(bot3, new Vec3(1,0,-3)) |
| | | ▪ | ▪ | ▪ | |
| Observation | First Person View | bot1 / bot2 / bot3 | bot1 / bot2 / bot3 | bot1 / bot2 | bot1 / bot2 / bot3 |
| | Inventory Information | bot1 has 5 bricks. bot1 has 2 sea_lantern... | bot1 has a stone_axe...bot3 has a stone_axe... | bot1 has 3 carrot. bot1 has 1 potato. bot2 has 3 carrot. bot2 has 2 beetroot... | bot1 has 1 beef ... bot3 has 1 iron_shovel |
| | Action | placeItem(bot1, 'coal_ore', new Vec3(0,1,-1)) placeItem(bot2, 'purple_wool', new Vec3(-1,1,0)) | mineBlock(bot1, new Vec3(0,0,1)) mineBlock(bot3, new Vec3(0,0,-2)) | farm_work(bot1, new Vec3(2,0,2), 'harvest') farm_work(bot2, new Vec3(1,0,-2), 'harvest') | takeOutFurnace(bot1, new Vec3(0,0,1)) |

Figure 5: Multi-modal prompt for Building, Clearing, Farming and Smelting.

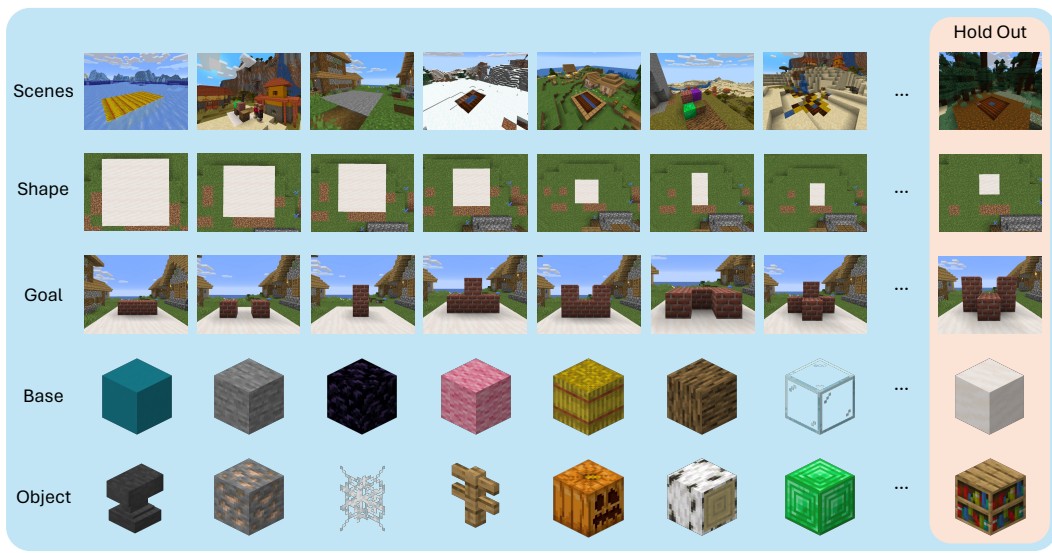

Figure 6: A close-up view of the visual diversity in tasks. The rightmost column displays the example holdout set for testing generalization.

# F  DATASET COMPONENT

The dataset is organized in the following structure. The folder "configure" contains the setup configurations and diversity settings for each task, with files named according to the task number. The folder "data" contains four sub-folders: sub-folders "1", "2", and "3" correspond to the first-person views of three different agents, while sub-folder "4" corresponds to the orthographic projections. Inside each of these sub-folders are screenshots for the respective agents, each labeled with a timestamp indicating the moment of each action. The folder "json" contains observation data for each agent, along with task-related information such as rewards, completion status ("done"), and timestamps.

```
task_building/
    |-- configure/
    |    |-- 0.json
    |    |-- 1.json
    |    |-- ...
    |-- data/
    |    |-- 0/
    |    |    |-- 1/
    |    |    |    |-- screenshot_<timestamp>.png
    |    |    |    |-- screenshot_<timestamp>.png
    |    |    |    |-- screenshot_<timestamp>.png
    |    |    |    |-- ...
    |    |    |-- 2/
    |    |    |    |-- screenshot_<timestamp>.png
    |    |    |    |-- ...
    |    |    |-- 3/
    |    |    |    |-- screenshot_<timestamp>.png
    |    |    |    |-- ...
    |    |    |-- 4/
    |    |    |    |-- screenshot_<timestamp>.png
    |    |    |    |-- ...
    |    |-- 1/
    |    |-- 2/
    |    |-- ...
    |-- json/
    |    |-- 0.json
    |    |-- 1.json
    |    |-- 2.json
    |    |-- ...
task_clearing/
    | ...
task_farming/
    | ...
task_smelting/
    | ...
```

## G   EXAMPLE TASK/DEMO

### G.1   GPT-4O PROMPT

You are controlling 3 bots in a Minecraft world. The goal is
    to build a specific structure on a platform.

    Please review the images provided below, which include the
        current state of the world and the goal structure (
        the final image is the three orthographic views of the
        goal). Based on these observations, generate actions
        for each bot to help build the structure.

    **Instructions:**

    – **Action Format:**

    – **Bots:**
    – `botID` can be one of: 'bot1', 'bot2', 'bot3', 'bot4' (
        depending on the number of bots).
    – **Blocks:**
    – `"block"` is the type of block to place.

```
    – **Available Blocks:**
    – 'oak_fence', 'birch_log', 'coal_ore', 'bricks', '
        sandstone', 'stone', 'iron_ore', 'gold_ore', 'sponge',
        'sea_lantern', 'dirt', 'grass_block', 'clay', '
        oak_planks', 'emerald_block', 'pumpkin', '
        orange_concrete', 'purple_wool', 'end_stone', '
        bookshelf', 'acacia_fence', 'oak_log'
    – **Constraints:**
    – **Inventory Awareness:** Ensure each bot has the
        necessary blocks in their inventory.
    – **No Overlapping Blocks:** Do not place more than one
        block at the same position.
    – **Workspace Dimensions:** The center of the workspace is
        at (0, 0, 0), and it spans 3 units along the x-axis,
        3 units along the z-axis, and 2 units along the y-axis
        .
    – **One Action per Bot:** Each bot can place only one
        block at a time.

    **Submission Guidelines:**

    – Provide only the list of action commands for all bots.
    – Do not include any additional text, explanations, or
        formatting (e.g., no code blocks or markdown).
    – Example:
    [ "placeItem(bot1, 'stone', new Vec3(1, 0, 0))", "
        placeItem(bot2, 'oak_planks', new Vec3(0, 0, 1))" ]
    You need to put "" each entry in the list.
    Please generate the list of commands based on the current
        observations and the goal image.
```

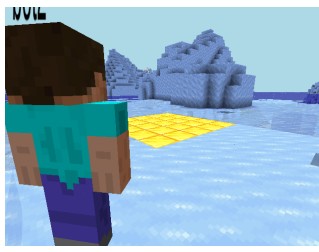

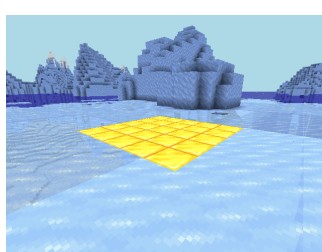

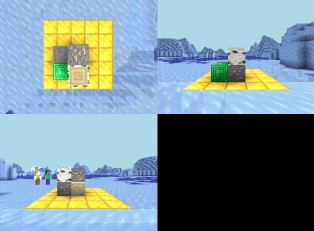

(a) Agent1 observation      (b) Agent2 observation      (c) three orthographic views of the goal

Figure 7: Observations to GPT-4o.

Additionally, we provide a one-shot example of the same task from the training set as an example to GPT-4o.

## G.2 GPT-4O COMMON ERRORS

We provide some errors from GPT-4 below:

Harvest without sow.

```
["farm_work(bot1, new Vec3(0, 0, 1), 'harvest')", "farm_work(
    bot2, new Vec3(0, 0, -1), 'harvest')"]
```

Sow never harvest.

```
["farm_work(bot1, new Vec3(1,0,1), 'sow', 'wheat')", "
    farm_work(bot2, new Vec3(1,0,0), 'sow', 'wheat')", "
    farm_work(bot3, new Vec3(0,0,0), 'sow', 'wheat')"]
```

Fail to understand 3D spatial relations.

```
[mineBlock(bot1, new Vec3(1,1,1))","mineBlock(bot2, new Vec3
    (-1,1,1))","mineBlock(bot3, new Vec3(-1,1,0))]
```

## H  DATASET STATISTICS TABLES

| Diversity | Type | Count | Percentage |
|---|---|---|---|
| **Action Sequences** | | | |
| | 3 | 7,777 | 51.85% |
| | 2 | 3,207 | 21.38% |
| | 4 | 3,091 | 20.61% |
| | 5 | 483 | 3.22% |
| | 6 | 440 | 2.93% |
| **Agents** | | | |
| | 3 | 7,505 | 50.03% |
| | 2 | 7,493 | 49.97% |
| **Scenes** | | | |
| | ice_on_water | 2,555 | 17.04% |
| | mountain_half | 2,553 | 17.03% |
| | village | 2,482 | 16.55% |
| | desert_village | 2,480 | 16.53% |
| | snow_mountain | 2,478 | 16.52% |
| | swamp | 2,450 | 16.34% |
| **Background Types** | | | |
| | stone | 1,530 | 10.20% |
| | pink_wool | 1,527 | 10.19% |
| | glowstone | 1,522 | 10.15% |
| | obsidian | 1,511 | 10.08% |
| | glass | 1,509 | 10.07% |
| | smooth_quartz | 1,499 | 10.00% |
| | hay_block | 1,494 | 9.96% |
| | gold_block | 1,473 | 9.82% |
| | oak_wood | 1,471 | 9.81% |
| | cyan_concrete | 1,462 | 9.75% |
| **Target Types** | | | |
| | bricks | 10,391 | 9.92% |
| | sponge | 5,438 | 5.19% |
| | coal_ore | 5,370 | 5.13% |
| | grass_block | 5,327 | 5.09% |
| | clay | 5,318 | 5.08% |
| | sea_lantern | 5,296 | 5.06% |
| | orange_concrete | 5,287 | 5.05% |
| | pumpkin | 5,269 | 5.03% |
| | purple_wool | 5,257 | 5.02% |
| | gold_ore | 5,247 | 5.01% |
| | oak_fence | 5,234 | 5.00% |
| | oak_planks | 5,216 | 4.98% |
| | birch_log | 5,184 | 4.95% |
| | stone | 5,182 | 4.95% |
| | sandstone | 5,176 | 4.94% |
| | emerald_block | 5,164 | 4.93% |
| | iron_ore | 5,160 | 4.93% |
| | dirt | 5,124 | 4.89% |
| | end_stone | 5,119 | 4.89% |

Table 6: Diversity Statistics for Task Building

| Diversity | Type | Count | Percentage |
|---|---|---|---|
| **Target Counts** | | | |
| | 6 | 5,653 | 37.69% |
| | 7 | 2,625 | 17.50% |
| | 8 | 2,573 | 17.15% |
| | 5 | 2,122 | 14.15% |
| | 10 | 526 | 3.51% |
| | 12 | 515 | 3.43% |
| | 9 | 496 | 3.31% |
| | 11 | 488 | 3.25% |
| **Dimensional Shapes** | | | |
| | [3, 1, 2] | 3,859 | 25.73% |
| | [4, 1, 2] | 3,770 | 25.14% |
| | [2, 3, 2] | 3,695 | 24.63% |
| | [2, 2, 2] | 3,674 | 24.49% |

Table 7: Diversity Statistics for Task Building (Cont.)

| Diversity | Type | Count | Percentage |
|---|---|---|---|
| **Action Sequences** | | | |
| | 4 | 4,027 | 27.51% |
| | 5 | 3,751 | 25.61% |
| | 6 | 3,270 | 22.32% |
| | 3 | 1,561 | 10.66% |
| | 7 | 1,396 | 9.53% |
| | 8 | 424 | 2.89% |
| | 9 | 133 | 0.91% |
| | 2 | 79 | 0.54% |
| **Agents** | | | |
| | 2 | 7,358 | 50.28% |
| | 3 | 7,283 | 49.72% |
| **Scenes** | | | |
| | desert_village | 3,012 | 20.56% |
| | snow_mountain | 2,948 | 20.13% |
| | swamp | 2,929 | 20.00% |
| | ice_on_water | 2,894 | 19.76% |
| | village | 2,858 | 19.54% |
| **Background Types** | | | |
| | smooth_quartz | 1,405 | 9.59% |
| | pink_wool | 1,357 | 9.27% |
| | gold_block | 1,353 | 9.24% |
| | oak_wood | 1,334 | 9.10% |
| | hay_block | 1,332 | 9.09% |
| | cyan_concrete | 1,332 | 9.09% |
| | grass_block | 1,328 | 9.06% |
| | glass | 1,325 | 9.04% |
| | glowstone | 1,309 | 8.93% |
| | stone | 1,302 | 8.89% |
| | obsidian | 1,264 | 8.63% |
| **Target Counts** | | | |
| | 6 | 4,310 | 29.43% |
| | 5 | 2,499 | 17.07% |
| | 4 | 2,436 | 16.64% |
| | 8 | 1,843 | 12.58% |
| | 7 | 1,803 | 12.31% |
| | 9 | 1,750 | 11.95% |
| **Target Types** | | | |
| | oak_fence | 5,879 | 6.45% |
| | grass_block | 5,836 | 6.40% |
| | clay | 5,816 | 6.38% |
| | oak_log | 5,772 | 6.33% |
| | sandstone | 5,748 | 6.30% |
| | acacia_fence | 5,744 | 6.30% |
| | birch_log | 5,732 | 6.28% |
| | bookshelf | 5,726 | 6.28% |
| | stone | 5,709 | 6.26% |
| | bricks | 5,695 | 6.25% |
| | crafting_table | 5,684 | 6.23% |
| | dirt | 5,671 | 6.22% |
| | cobweb | 5,605 | 6.15% |
| | iron_ore | 5,603 | 6.14% |
| | coal_ore | 5,555 | 6.09% |
| | anvil | 5,439 | 5.96% |
| **Dimensional Shapes** | | | |
| | 3 | 7,346 | 50.15% |
| | 2 | 7,295 | 49.84% |

Table 8: Diversity Statistics for Task Clearing

| Diversity | Type | Count | Percentage |
|---|---|---|---|
| **Tools** | | | |
| | stone_pickaxe | 9,329 | 25.51% |
| | stone_sword | 9,180 | 25.10% |
| | stone_axe | 9,150 | 24.99% |
| | stone_shovel | 8,906 | 24.36% |
| **Dimensional Shapes** | | | |
| | 3 | 7,346 | 50.15% |
| | 2 | 7,295 | 49.84% |

Table 9: Diversity Statistics for Task Clearing (Cont.)

| Diversity | Type | Count | Percentage |
|---|---|---|---|
| **Action Sequences** | | | |
| | 4 | 7,458 | 50.33% |
| | 5 | 3,731 | 25.17% |
| | 3 | 3,264 | 22.02% |
| | 6 | 270 | 1.82% |
| | 2 | 81 | 0.55% |
| | 7 | 11 | 0.07% |
| **Agents** | | | |
| | 2 | 7,465 | 50.37% |
| | 3 | 7,350 | 49.63% |
| **Scenes** | | | |
| | snow_mountain | 3,732 | 25.18% |
| | swamp | 3,722 | 25.11% |
| | ice_on_water | 3,707 | 25.01% |
| | village | 3,654 | 24.69% |
| **Background Types** | | | |
| | stone | 2,892 | 19.51% |
| | obsidian | 1,549 | 10.46% |
| | hay_block | 1,527 | 10.30% |
| | oak_wood | 1,524 | 10.28% |
| | cyan_concrete | 1,492 | 10.06% |
| | glass | 1,465 | 9.88% |
| | smooth_quartz | 1,462 | 9.86% |
| | pink_wool | 1,455 | 9.81% |
| | dirt | 1,449 | 9.77% |
| **Target Types** | | | |
| | potato | 4,972 | 33.56% |
| | carrot | 4,955 | 33.45% |
| | wheat | 4,888 | 32.99% |
| **Target Counts** | | | |
| | 4 | 2,873 | 19.39% |
| | 3 | 2,269 | 15.31% |
| | 5 | 2,256 | 15.22% |
| | 6 | 2,151 | 14.51% |
| | 2 | 1,240 | 8.37% |
| | 8 | 1,112 | 7.50% |
| | 10 | 1,062 | 7.17% |
| | 7 | 933 | 6.29% |
| | 12 | 512 | 3.45% |
| | 14 | 407 | 2.75% |

Table 10: Diversity Statistics for Task Farming

| Diversity | Type | Count | Percentage |
|---|---|---|---|
| **Action Sequences** | | | |
| | 5 | 3,261 | 30.20% |
| | 4 | 3,072 | 28.45% |
| | 6 | 2,041 | 18.89% |
| | 3 | 1,824 | 16.88% |
| | 2 | 358 | 3.31% |
| | 7 | 239 | 2.21% |
| | 8 | 8 | 0.07% |
| **Agents** | | | |
| | 3 | 5,480 | 50.75% |
| | 2 | 5,323 | 49.25% |
| **Scenes** | | | |
| | snow_mountain | 2,272 | 21.04% |
| | desert_villege | 2,257 | 20.92% |
| | swamp | 2,171 | 20.08% |
| | ice_on_water | 2,059 | 19.09% |
| | villege | 2,044 | 18.87% |
| **Background Types** | | | |
| | gold_block | 1,014 | 9.22% |
| | smooth_quartz | 1,010 | 9.19% |
| | cyan_concrete | 995 | 9.02% |
| | glowstone | 981 | 8.92% |
| | pink_wool | 990 | 8.99% |
| | glass | 978 | 8.89% |
| | oak_wood | 987 | 8.98% |
| | grass_block | 977 | 8.88% |
| | hay_block | 968 | 8.80% |
| | stone | 964 | 8.76% |
| | obsidian | 939 | 8.54% |
| **Furnace** | | | |
| | 1 | 5,772 | 53.45% |
| | 2 | 5,031 | 46.55% |
| **Fuel Types** | | | |
| | coal_block | 999 | 9.58% |
| | charcoal | 962 | 9.22% |
| | lava_bucket | 940 | 9.01% |
| | coal | 921 | 8.84% |
| | spruce_planks | 910 | 8.73% |
| | acacia_planks | 906 | 8.69% |
| | oak_planks | 861 | 8.26% |
| | birch_log | 893 | 8.57% |
| | acacia_log | 887 | 8.50% |
| | spruce_log | 845 | 8.10% |
| | oak_log | 840 | 8.05% |
| | birch_planks | 839 | 8.04% |

Table 11: Diversity Statistics for Task Smelting

| Diversity | Type | Count | Percentage |
|---|---|---|---|
| **Goal Types** | | | |
| | food | 5,412 | 50.09% |
| | item | 5,391 | 49.91% |
| **Target Types** | | | |
| | glass | 1,144 | 10.26% |
| | gold_ingot | 1,094 | 9.81% |
| | stone | 1,077 | 9.66% |
| | smooth_sandstone | 1,040 | 9.32% |
| | iron_ingot | 1,036 | 9.29% |
| | cooked_salmon | 712 | 6.38% |
| | cooked_cod | 708 | 6.35% |
| | baked_potato | 758 | 6.80% |
| | cooked_mutton | 664 | 5.95% |
| | cooked_rabbit | 648 | 5.81% |
| | cooked_porkchop | 668 | 5.99% |
| | cooked_beef | 627 | 5.62% |
| | cooked_chicken | 627 | 5.62% |
| **Target Counts** | | | |
| | 2 | 3,999 | 37.01% |
| | 3 | 3,363 | 31.13% |
| | 1 | 1,909 | 17.68% |
| | 4 | 1,532 | 14.18% |
| **Tools** | | | |
| | iron_pickaxe | 18,633 | 29.69% |
| | iron_shovel | 13,676 | 21.78% |
| | iron_axe | 13,453 | 21.43% |
| | iron_sword | 13,448 | 21.42% |
| **Resource Types** | | | |
| | red_sand | 2,032 | 10.37% |
| | gold_ore | 1,999 | 10.20% |
| | cobblestone | 1,915 | 9.77% |
| | sandstone | 1,818 | 9.28% |
| | iron_ore | 1,780 | 9.08% |
| | coal_ore | 1,714 | 8.75% |
| | acacia_planks | 1,564 | 7.98% |
| | oak_planks | 1,503 | 7.67% |
| | birch_log | 1,486 | 7.58% |
| | spruce_log | 1,477 | 7.54% |
| | oak_log | 1,456 | 7.44% |
| | spruce_planks | 1,471 | 7.51% |
| | birch_planks | 1,344 | 6.86% |
| | sheep | 1,119 | 5.71% |
| | pig | 1,104 | 5.63% |
| | rabbit | 1,097 | 5.60% |
| | chicken | 1,081 | 5.52% |
| | cow | 700 | 3.57% |
| | mooshroom | 675 | 3.44% |

Table 12: Diversity Statistics for Task Smelting (Cont.)

