# OpenReview forum: "TeamCraft: A Benchmark for Embodied Multi-Agent Systems in Minecraft"
_ICLR.cc/2025/Conference — ICLR 2025 Conference Withdrawn Submission_

### Official Review · Reviewer_BzTY · 2024-10-21

**Soundness:** 1
**Presentation:** 1
**Contribution:** 1
**Rating:** 3
**Confidence:** 5

**Summary:**

The paper introduces TeamCraft, a new benchmark designed to advance research in embodied multi-agent (MA) systems using the Minecraft environment. Unlike traditional MA research that relies on abstract vector inputs, TeamCraft provides agents with multi-modal task specifications and observations, including three orthographic views of the environment and language instructions. Agents are tasked with collaborating efficiently to complete assignments such as building construction, clearing areas, farming, and smelting within diverse, procedurally-generated environments. The benchmark includes a dataset of 70,000 procedurally-generated demonstrations featuring over 50 objects across various scenes. The authors also explore the application of GPT-4o and LLAVA models tailored to multi-agent scenarios, evaluating several Vision-Language Model (VLM) multi-agent control strategies in both centralized and decentralized settings. The TeamCraft platform and dataset are made publicly available to the research community.

**Strengths:**

1. Introduction of a Novel Benchmark: TeamCraft offers a new and challenging benchmark that fills the MA research community gap by providing a platform that emphasizes realistic, multi-modal inputs in complex 3D environments.
2. Rich Multi-Modal Dataset: Creating a large-scale dataset with 70,000 procedurally generated demonstrations is a significant contribution, providing valuable resources for training and evaluating MA systems.
3. Emphasis on Realistic Agent Inputs: By moving away from abstract vector inputs and providing agents with visual and language inputs, the benchmark encourages the development of agents that can handle more realistic and complex sensory inputs.
4. Application of VLMs in MA Systems: The exploration of GPT-4o and LLaVA models in multi-agent scenarios showcases innovative applications of Vision-Language Models in the context of MA control strategies.
5. Open-Source Contribution: Making the TeamCraft platform and dataset publicly available promotes transparency and enables other researchers to build upon this work, fostering collaboration within the community.

**Weaknesses:**

1. Weak Experimental Choices: The selected tasks, building, clearing, farming, and smelting, are similar and primarily involve adding or deleting blocks in a small scene. This similarity raises concerns about whether these tasks truly necessitate a multi-agent system. Dividing such similar tasks into four distinct categories may not effectively demonstrate the advantages of a multi-agent framework. The absence of more diverse tasks, such as general navigation that requires long-path planning, limits demonstrating the multi-agent system’s capabilities.
2. Inadequate Benchmarking: While the benchmark is presented as a main contribution, detailed comparison standards and scoring methods are lacking. It’s unclear what metrics are used to evaluate the models, whether these methods are reasonable, and if their validity has been verified. The baseline comparison is also insufficient, relying only on Vicuna. A robust benchmark should involve multiple language models to provide a comprehensive evaluation.
3. Limited Novel Contribution: The application of GPT-4o and LLAVA models to multi-agent scenarios is insufficient as a standalone innovative contribution. Applying existing models without significant modification or novel integration does not strongly advance the field.

**Questions:**

1. Necessity of a Multi-Agent System for Similar Tasks: Given that the tasks you selected are similar and involve straightforward actions like adding or deleting blocks in a confined space, do these tasks genuinely require a multi-agent system? Could you elaborate on why these tasks were chosen and divided into four different categories?
2. Choice of Tasks Over General Navigation Tasks: Since the goal is to demonstrate multi-agent performance, why not include more popular and general navigation tasks that require long-path planning and could better highlight the advantages of a multi-agent framework?
3. Benchmark Evaluation Methods: What calculation standards and scoring methods are used to evaluate the models in your benchmark? Are these methods reasonable, and have they been validated? Additionally, why does the benchmark only include Vicuna as the baseline? Would incorporating more language models provide a more thorough and complete comparison?
4. Clarification on Innovative Contributions: What novel contributions does your work offer beyond applying GPT-4o and LLaVA models to multi-agent scenarios? How does your approach go beyond simply using existing models in a new context to provide meaningful advancements in the field?

---

### Official Review · Reviewer_A2ik · 2024-11-03

**Soundness:** 2
**Presentation:** 1
**Contribution:** 2
**Rating:** 3
**Confidence:** 4

**Summary:**

This paper presents TeamCraft, a multi-agent benchmark based on Minecraft. The benchmark provides each agent with multi-modal image and language observations, and the agents need to coordinate and collaborate together to solve the tasks. ~100k demonstrations featuring ~50 objects are also provided. Finally, the authors provide multi-agent LLaVa and GPT-4o baselines.

**Strengths:**

Multi-agent collaboration is an important research problem, with algorithms applicable in a wide range of real-world scenarios. The benchmark provides a suitable platform for studying multi-agent collaboration, particularly with multi-modal vision and language inputs.

**Weaknesses:**

- The contribution of this work seems quite marginal compared to the previous works. According to Table 1, the multi-agent decentralized aspect of the benchmark is not novel, as MindLand (Gong et al., published in 2023) has already presented a similar benchmark. Instead, it seems that a valuable contribution of the benchmark is the 3D relationship reasoning and tool use ability (when compared with prior work) on top of multi-agent planning, but these aspects are neither described in detail, nor emphasized in the paper. Furthermore, does multi-agent 3D relationship reasoning and multi-agent tool use present unique challenges compared to the single-agent counterparts? If so, this should be emphasized and analyzed in detail in the paper.
- Most of the introduction and related work is clearly written or at least heavily edited by GPT-4 / 4o (e.g., repeated use of "encompass", "foster", "employ", and "subsequent"; "culminating in the prediction of ultimate actions" (L57); "decidedly more formidable and challenging" (L43); "It utilizes the acclaimed Minecraft game as an experimental platform"; "each assignment necessitates consideration of fellow agents, spanning factors such as ..."; "comprehensively encodes input facets and subsequently fuses embeddings" (L56)). I'm not sure how serious of an issue this is; it will leave up to ACs to review and decide.

**Questions:**

- For the proposed baselines, the GPT-4o method significantly underperforms multi-agent LLaVa. It seems that this is because GPT-4o does not have a prior understanding of the complete rules of Minecraft. If rules and constraints in Minecraft are provided as part of the prompt (e.g., one can only harvest after sow), will it perform much better?

---

### Official Review · Reviewer_UVJG · 2024-11-04

**Soundness:** 2
**Presentation:** 3
**Contribution:** 3
**Rating:** 6
**Confidence:** 4

**Summary:**

The paper introduces TeamCraft, a multi-agent,  multi-modal benchmark platform based on Minecraft. TeamCraft features 4 main task categories and includes 70,000 procedurally-generated demonstrations across varied environments.

**Strengths:**

1. The paper is well-written and easy to understand.
2. TeamCraft is more comprehensive than previous benchmarks together with substantial demonstrations.

**Weaknesses:**

1. The experimental evaluation suffers from a narrow selection of models:
- Only LLaVa series models and GPT-4o are evaluated.
- The paper could have included other VLMs to further prove the robustness of the benchmark.


2. While the paper presents a multi-agent benchmark, the analysis of multi-agent interactions is superficial:
- Limited analysis of how agents coordinate and communicate
- No detailed analysis of how different numbers of agents affect performance

**Questions:**

See weaknesses.

---

### Official Review · Reviewer_6F4W · 2024-11-04

**Soundness:** 1
**Presentation:** 1
**Contribution:** 1
**Rating:** 1
**Confidence:** 3

**Summary:**

This paper presents a new benchmark for multi-agent systems used for Minecraft. The key to team craft is to provide agents with multi-modal task specifications and observation. The LLaVA and GPT-4o is used for evaluation on the proposed benchmark.

**Strengths:**

I can't identify any clear strengths in this paper.

**Weaknesses:**

There is no demo available for this paper, and the link provided in the abstract is empty. Only a few samples are included in the Appendix. The quality of the provided data is significantly lower than that of MineDojo in terms of data diversity, despite the authors' claim of having over 70,000 trajectories for evaluating model generalization. This makes it difficult to assess the quality of the benchmark.

Based on Table 1, this new benchmark appears to be incremental compared to prior work. The choice of comparison targets is also unusual. SQA3D is designed for 3D QA evaluation, and both Alexa Arena and OpenEQA are used for QA assessment in indoor scenes, so it is confusing that the authors chose to include these datasets in a comparative table.

It is also unclear how the pretrained LLM is used for evaluation. LLaVA is not specifically trained to play Minecraft, and even though the model is fine-tuned, only one open-source LLM is evaluated. The prompt for GPT-4 is also missing. Additionally, there are several works, such as Voyager[1] and GITM[2], that use agents to play Minecraft. For a benchmark, a comprehensive evaluation of existing popular methods is essential.

[1] VOYAGER: An Open-Ended Embodied Agent with Large Language Models
[2] Ghost in the Minecraft: Generally Capable Agents for Open-World Environments via Large Language Models with Text-based Knowledge and Memory

**Questions:**

Please refer to the weaknesses section. This paper appears to be half-finished. Although I’m not an expert in this domain, there are numerous flaws: insufficient data examples, no video demo, no description of the benchmark, no analysis of related work in this area, and no detailed comparison with existing benchmarks. Overall, this paper falls significantly below the standards expected for this conference.

---

### Note · Authors · 2024-11-13

I have read and agree with the venue's withdrawal policy on behalf of myself and my co-authors.